

# Trainee psychotherapists' emotion recognition accuracy during 1.5 years of psychotherapy education compared to a control group: no improvement after psychotherapy training

Lillian Döllinger[1], Isabelle Letellier[1,2], Lennart Högman[1], Petri Laukka[1], Håkan Fischer[1] and Stephan Hau[1]

[1] Department of Psychology, Stockholm University, Stockholm, Sweden
[2] Department of Child and Youth Studies, Stockholm University, Stockholm, Sweden

## ABSTRACT

The ability to recognize and work with patients' emotions is considered an important part of most psychotherapy approaches. Surprisingly, there is little systematic research on psychotherapists' ability to recognize other people's emotional expressions. In this study, we compared trainee psychotherapists' nonverbal emotion recognition accuracy to a control group of undergraduate students at two time points: at the beginning and at the end of one and a half years of theoretical and practical psychotherapy training. Emotion recognition accuracy (ERA) was assessed using two standardized computer tasks, one for recognition of dynamic multimodal (facial, bodily, vocal) expressions and one for recognition of facial micro expressions. Initially, 154 participants enrolled in the study, 72 also took part in the follow-up. The trainee psychotherapists were moderately better at recognizing multimodal expressions, and slightly better at recognizing facial micro expressions, than the control group at the first test occasion. However, mixed multilevel modeling indicated that the ERA change trajectories for the two groups differed significantly. While the control group improved in their ability to recognize multimodal emotional expressions from pretest to follow-up, the trainee psychotherapists did not. Both groups improved their micro expression recognition accuracy, but the slope for the control group was significantly steeper than the trainee psychotherapists'. These results suggest that psychotherapy education and clinical training do not always contribute to improved emotion recognition accuracy beyond what could be expected due to time or other factors. Possible reasons for that finding as well as implications for the psychotherapy education are discussed.

Corresponding author
Lillian Döllinger,
lillian.dollinger@psychology.su.se

## INTRODUCTION

Recognizing other people's emotions is a central part of all interpersonal communication. In psychotherapy in particular, it is essential for psychotherapists to be able to correctly identify and work with patients' emotional expressions (*Greenberg & Safran, 1989*). Beyond the verbal display of emotion, nonverbal displays have gained more and more attention in research. People can express emotions with their body posture, their facial expressions, and their tone of voice, both on a conscious and an unconscious level. Being able to read those nonverbal displays of emotion provides important information about patients' affective states (*Greenberg & Safran, 1989*) and can be considered an essential part of psychotherapists' and other mental health care providers' work. Interventions that focus on emotions, like exposure to and processing of emotions in session, have been found to be beneficial for patients' psychotherapy results (see, *e.g.*, *Greenberg & Safran, 1989*; *Watson, 2018*). Several studies indicate that psychotherapists' empathy (see, *e.g.*, *Elliott et al., 2011*, *2018*; *Nienhuis et al., 2018*; *Wampold, 2015*) and emotion recognition accuracy (ERA; see, *e.g.*, *Abargil & Tishby, 2021*) are good predictors for psychotherapy outcome. The present study is focusing on trainee psychotherapists' ERA, as it can be understood as the perceptive or perspective-taking component of empathy (*reading* someone else's emotion), in contrast to the affective or feeling-component of empathy (*feeling* someone else's emotion) (see, *e.g.*, *Martingano & Konrath, 2022*). It is also plausible that ERA is a prerequisite for successful empathy (see, *e.g.*, *Besel & Yuille, 2010*).

Although most psychotherapists would agree that the work with emotions and patients' nonverbal emotional expressions is important and there are studies linking high ERA and empathy to positive psychotherapy results, there is little systematic research on psychotherapists' ERA. Most publications in the field consist of comparisons of psychotherapists to other groups (*e.g.*, other professions or undergraduate students), hypothesizing that psychotherapists have superior ERA based on their personality traits, interest for others, clinical training or simply due to experience of regularly working with patients. Others are studying the impact of experience and clinical training on ERA (*e.g.*, comparing experienced professionals and trainees). Empirically, there are mixed results. *Hutchison & Gerstein (2012)*, for example, found no difference in nonverbal ERA between counseling trainees and undergraduate students using an ERA task following Ekman's (see, *e.g.*, *Ekman & Cordaro, 2011*) theory of basic emotions that consisted of still pictures of facial expressions from the *Japanese and Caucasian Facial Expression of Emotion* database (*Matsumoto & Ekman, 1988*). Similarly, *Hassenstab et al. (2007)* did not find a difference between a small sample of practicing psychotherapists and matched controls when it comes to detecting facial expressions of emotion (although, they found them to be superior when it comes to making inferences based on language). Similarly, this task also used still pictures of faces from one of Ekman's databases (*Ekman & Friesen, 1971*). These findings speak against the hypothesis that psychotherapists possess superior ERA for nonverbal displays. In contrast, *Pauza et al. (2010)* found that psychotherapy trainees in the beginning of their education (irrespective of their therapeutic orientation) were superior to coaches in the beginning of their education, a normal population sample and patients with

anxiety disorders. Also this study used the *Japanese and Caucasian Facial Expression of Emotion* database (*Matsumoto & Ekman, 1988*) for their ERA task. *Machado, Beutler & Greenberg (1999)*, in contrast to the other named studies, compared verbal and nonverbal ERA in experienced psychotherapists and undergraduate psychology students who intended to become psychotherapists using a video recording of a therapy session (naturalistic), an altered version in which the verbal content was filtered out and no longer identifiable (nonverbal), and a written transcript of the session (verbal). They found that the psychotherapists were much more accurate in identifying nonverbal emotional expressions than the undergraduate students, who seemed to rely more on a combination of verbal and nonverbal input, concluding that the amount of clinical experience and training enhances (nonverbal) ERA. These studies are in favor of the assumption that psychotherapists have greater ERA than other populations. When investigating other mental health professionals (like psychiatrists or nurses), two studies (*Arango de Montis et al., 2014*; *Minardi, 2013*) found some indication to believe that clinical training or experience leads to increased ERA, whereas another study (*Ragsdale et al., 2016*) did not find a difference in ERA between medical faculty and medical students. To sum up, the research about psychotherapists' ERA in comparison to other populations, and about whether ERA improves due to clinical training and experience, is inconclusive and sample sizes are often small. Still, research with larger samples (*Pauza et al., 2010*) or more ecologically valid measures (*Machado, Beutler & Greenberg, 1999*) tends to support the idea that psychotherapists might possess higher ERA and that ERA increases with clinical experience.

## Present study

The present study aimed to deepen the knowledge about psychotherapists' ERA by investigating trainee psychotherapists both at the beginning and the end of theoretical and practical psychotherapy training, and comparing their results on a variety of ERA variables to results from a control group of undergraduate students. We chose to investigate two facets of ERA, namely the recognition accuracy for emotions in multiple modalities and the recognition accuracy for very brief facial expressions, so called micro expressions of emotion (*Ekman & Friesen, 1969*). Emotions are not only communicated *via* facial expressions, but also *via* bodily postures and tone of voice, among other channels (see, *e.g.*, *Bänziger, 2016*; *Bhatara, Laukka & Levitin, 2014*; *de Gelder, de Borst & Watson, 2015*). For this reason, it is relevant to investigate ERA in a more ecologically valid fashion, using an ERA task that includes facial, bodily and nonverbal vocal cues, and *via* dynamic stimuli. Further, the assessment of single modalities (audio-only, video-only) could provide insights into whether there are specific modalities that trainee psychotherapists are particularly skilled in or might need targeted training for. We suggest that the investigation of micro expressions is very relevant for the psychotherapy context as well, as the recognition of very brief emotional expressions could potentially lead to relevant insights related to patients' mental states, like mixed, hidden, repressed, denied or dissociated affects, and to using appropriate interventions (see, *e.g.*, *Ekman & Friesen, 1969*).

We expected the trainee psychotherapists to have greater multimodal and micro expression ERA than the undergraduate student population, possibly due to a heightened interest in socio-emotional communication and a stronger focus on their interaction partners' mental states. We were further interested in whether the trainee psychotherapists' multimodal and micro expression ERA increased more over time compared to the control group, possibly by receiving specialized practical and theoretical psychotherapy training and treating patients at the university clinic. The use of video-feedback methods in supervision, and knowledge about and attention towards working with emotions in psychotherapy, could potentially act as a training *per se*. To summarize, our hypotheses were (a) that trainee psychotherapists' ERA at baseline would be higher than the control group's ERA (*hypothesis 1*), and (b) that the trainee psychotherapists' ERA would increase more over time than the control group's ERA (*hypothesis 2*).

# MATERIALS AND METHODS

## Participants and data collection

Initially, 154 healthy participants were enrolled in the study, consisting of 49 trainee psychotherapists and 105 undergraduate students. See Table 1 for descriptive statistics and statistical comparisons of the two groups at the different time points. The age difference between the trainee psychotherapists ($M = 30.35$, $SD = 7.15$, range = 22–50) and the control group ($M = 25.82$, $SD = 7.78$, range = 18–61) at pretest was significant and of moderate magnitude; there were no gender differences. The trainee psychotherapists were enrolled in Stockholm university's clinical psychology training program; 34 trainee psychotherapists studied psychodynamic psychotherapy (PDT) and 15 studied cognitive behavioral therapy (CBT). The control group attended other courses or programs at the university and consisted of mostly psychology undergraduate students, students in work and organizational psychology and students of related subjects like sociology and anthropology. None had received any clinical training. The number of participants was not specified in advance. Instead, we tried to include as many trainee psychotherapists as possible out of three cohorts of students starting their practical training. We aimed to collect about twice as much data for the control group since experience shows that this group of students has a higher dropout rate because many only attend a few classes in psychology and then move on to other institutions. Data collections for the trainee psychotherapists and the control group ran concurrently. The participants were recruited *via* posting boards, email lists and oral presentations of the project. The participants received course credits or movie tickets for their participation and during lunch hours also a sandwich.

Data was collected at two time points: in the beginning of the trainee psychotherapists' practical work with patients at the university clinic (term 7; pretest) and then again in the end of clinical training, which lasted about one and a half years (term 9; follow-up). At follow-up, 72 participants completed data collection a second time (31 trainee psychotherapists and 41 undergraduate students). The control group was still significantly younger and there were still no gender differences between the groups (see Table 1).
**Table 1 Sample characteristics: descriptive statistics (means, standard deviations, range, count, 95% confidence intervals) and group comparisons (two-sided Student's *t*-test and two-sided Wilcoxon-Mann-Whitney *U*-test).**

| Measures | Trainee psychotherapists M (SD) range | Control group M (SD) range | Total M (SD) range | Statistic t/U [95% CI] | Effect size d/r [95% CI][b] |
|---|---|---|---|---|---|
| **Age** | | | | | |
| Pre (N = 154) | 30.35 (7.15) 22–50 (n = 49) | 25.82 (7.78) 18–61 (n = 105) | 27.26 (7.85) 18–61 | U = 3,774 p < 0.001*** [3.00–6.00] | r = 0.38 [0.25–0.51] |
| Follow up (n = 72) | 31.77 (7.62) 22–50 (n = 31) | 28.00 (9.79) 18–61 (n = 41) | 29.62 (9.06) 18–61 | U = 872 p < 0.01** [1.00–8.00] | r = 0.32 [0.11–0.52] |
| | **Count** | **Count** | **Count** | **t/U [95% CI]** | **d/r [95% CI][b]** |
| **Gender** | | | | | |
| Pre (N = 154) | 34 women, 15 men (n = 49) | 78 women, 27 men (n = 105) | 112 women, 42 men | U = 2,698 p = 0.53 [−0.00 to 0.00] | r = 0.05 [0.00–0.23] |
| Follow up (n = 72) | 19 women, 12 men (n = 31) | 31 women, 10 men (n = 41) | 50 women, 22 men | U = 726 p = 0.20 [−0.00 to 0.00] | r = 0.15 [0.01–0.36] |
| **Therapy approch** | | | | | |
| Pre (n = 49) | PDT = 34 CBT = 15 | | | | |
| Follow up (n = 31) | PDT = 20 CBT = 11 | | | | |
| | **M (SD) [95% CI]** | **M (SD) [95% CI]** | **M (SD) [95% CI]** | **t/U [95% CI]** | **d/r [95% CI][b]** |
| **State affect (PANAS)** | | | | | |
| Pre: positive (N = 154) | 3.11 (0.55) [2.96–3.27] (n = 49) | 3.22 (0.58) [3.11–3.34] (n = 105) | 3.19 (0.57) [3.10–3.28] | t(98.6) = −1.13 p = 0.26 [−0.30 to 0.08] | d = −0.19 [−0.51 to 0.13] |
| Pre: negative (N = 154) | 1.39 (0.36) [1.29–1.50] (n = 49) | 1.39 (0.35) [1.32–1.46] (n = 105) | 1.39 (0.35) [1.33–1.45] | U = 2,570 p = 0.99 [−0.10 to 0.10] | r = 0.00 [0.00–0.19] |
| Follow-up: positive (n = 45) | 3.01 (0.58) [2.84–3.17] (n = 17) | 3.00 (0.73) [2.86–3.14] (n = 28) | 3.00 (0.63) [2.82–3.19] | t(43) = 0.04 p = 0.97 [−0.39 to 0.40] | d = 0.01 [−0.67 to 0.7] |
| Follow-up: negative (n = 47) | 1.34 (0.38) [1.24–1.45] (n = 18) | 1.36 (0.45) [1.27–1.44] (n = 29) | 1.35 (0.4) [1.23–1.47] | U = 269 p = 0.87 [−0.10 to 0.10] | r = 0.03 [0.01–0.34] |

**Note:**
[95% CI][b]: 95% Confidence Interval for effect size is based on 1,000 bootstrap resamples of the mean difference (percentile interval). Common standardized effect size estimates: Cohen's *d*: $d = 0.2$ (small effect), $d = 0.5$ (moderate effect) and $d = 0.8$ (large effect); Wilcoxon-Mann-Whitney *U*-test: $r = 0.10$ (small), $r = 0.30$ (moderate), $r = 0.50$ (large). **$p < 0.01$, ***$p < 0.001$ (Holm adjusted).

Eighty-two participants had dropped out, we do not have information as to why this occurred.

## Psychotherapy education

The psychotherapy education part of the clinical psychology program at Stockholm university starts in term 6 in which the students receive introductions to PDT and CBT

and learn the basics of both therapy approaches. Then, they chose which approach they want to train more in depth, and practice at the university clinic from term 7 to the end of term 9 (about 1.5 years of specified clinical training). In the PDT group, the students learn about the basics of psychodynamic theory and method (*e.g.*, establishing a psychodynamic situation, the therapeutic frame, transference and countertransference, defense) and about contemporary concepts in PDT (*e.g.*, attachment theory, mentalizing), about how to work with patients with different levels of psychic integration, about developmental models and about psychodynamic diagnosis. They also learn about specific and short-term psychodynamic approaches, *e.g.*, *Dynamic Interpersonal Therapy* (*Lemma, Target & Fonagy, 2011*) and *Intensive Short-Term Psychodynamic Psychotherapy* (*Davanloo, 2001*). The CBT group learns about diagnosis, theories and methods within CBT (*e.g.*, case formulation, exposure and behavioral experiments, skills training, cognitive techniques, emotion regulation) and also about specific treatments for different diagnoses (*e.g.*, anxiety-related disorders, depression, personality disorders), like *Unified Protocol* (*Barlow et al., 2011*) or *Dialectical Behavior Therapy* (*Linehan, 2014*). They also learn about generic therapeutic skills, like establishing a therapeutic alliance.

The psychotherapy education for both groups consists of lectures, seminars and practical exercises (*e.g.*, filmed roleplays and exercises) and the CBT group even uses deliberate practice of certain skills. During those 1.5 years, the PDT and CBT students practice psychotherapy under supervision at the university clinic. The PDT students generally conduct one long-term psychotherapy and one short-term therapy (Dynamic Interpersonal Therapy) and the CBT students conduct at least one specific phobia treatment and then several treatments of patients with other diagnoses, like anxiety and depression. In supervision, they receive guidance regarding specific therapeutic methods and techniques, but also for developing generic therapist skills and traits, like empathy, self-reflection and other competencies.

In both approaches, as they are taught at Stockholm university, working with patients' emotions is an integral part of therapy. In the PDT classes, the trainees discuss patients' experiences and expressions of emotion in case formulations and treatment plans, particularly regarding emotions that are defended against (*e.g.*, in terms of Malan's triangle of conflict; *Malan, 1995*), or, *e.g.*, when discussing mentalizing capacities. In the CBT education, there are several course moments in which the trainees focus on trying to identify and analyze patients' emotions, *e.g.*, in case formulations, functional analysis of behavior and treatment plans. Practical exercises for communication skills also include nonverbal communication. Further, in role plays, supervision and in the training therapies, both CBT and PDT trainees frequently reflect about their patients' emotions and related concepts (*e.g.*, emotion regulation, experiencing of emotions in the body, empathy). They also learn to reflect about, handle or make use of their own emotional reactions (*e.g.*, becoming aware of their own emotions, understanding and using countertransference). No explicit, standardized ERA training as used in ERA research (*e.g.*, in the form of computerized trainings; see, *e.g.*, *Döllinger et al., 2021*; *Rebeschini et al., 2019*; *Schlegel et al., 2017*) is facilitated in either part of the program.

## Procedure

Both lab sessions (pretest and follow-up) started with reporting state affectivity before conducting two computerized ERA tasks. First, the participants did a test about classifying very brief facial emotional expressions, so called micro expressions (see *Ekman, 2003*; *Ekman & Friesen, 1969*; *Matsumoto & Hwang, 2018*), and then a test about classifying emotional expressions in several modalities (see *Material*). At pretest, all participants also filled in a questionnaire about adult attachment and the trainee psychotherapists also answered questionnaires about attitudes regarding psychotherapy, empathy, compassion and self-compassion (see Supplemental Material). The study was approved by the Swedish Ethical Review Authority (*Etikprövningsmyndigheten*, dnr 2014/1265-31) and all participants gave their written informed consent prior to participation.

## Material

We assessed nonverbal ERA in multiple modalities using the *Emotion Recognition Assessment in Multiple Modalities* test (ERAM; *Laukka et al., 2021*). This 72-item computerized task provides the possibility to assess ERA separately for dynamic video, audio and audio-video items taken from the *Geneva Multimodal Emotion Portrayals* (GEMEP; *Bänziger, Mortillaro & Scherer, 2012*). The ERAM test includes video and/or audio recordings of actors portraying twelve emotions (*anger, anxiety, despair, disgust, fear, interest, joy, pleasure, pride, relief, irritation*, and *sadness*), and the participants judge which emotion was expressed by choosing one out of 12 response alternatives (same as the included emotions). Based on the modality, the participants are presented with either only facial expressions and some body movement (video modality), only prosody using a pseudo language (*e.g.*, "ne kal i bam sud molen"; audio modality) or a combination of those two modalities (audio-video modality). There is also the opportunity to group the emotions according to their valence (positive, negative) and arousal (high, low) level (see *Bänziger, Mortillaro & Scherer, 2012*). The main measure for emotions in multiple modalities is the total score of all 72 items (ERAM total), but we also report ERA results for the separate modalities and for the valence and arousal categories. The task was conducted in Swedish. Internal consistency was assessed *via* Cronbach's alpha (*Cronbach, 1951*) and McDonald's omega coefficient (ω; *McDonald, 1999*). Omega hierarchical ($\omega_h$) is an estimate of reliability that is used to describe general factor saturation or the proportion of variance that can be attributed to one underlying general factor—in our case ERA. Omega total ($\omega_t$) is an estimate that is used to describe the proportion of variance that is due to all latent factors, meaning all subscales as well as a general factor—in our case the twelve emotions and ERA as general factor (see *McDonald, 1999*; *McNeish, 2018*; *Revelle, 2009*; *Revelle & Zinbarg, 2009*). In the present study, the alpha values for the pretest and follow-up were $\alpha_{pre} = 0.62$ and $\alpha_{follow-up} = 0.59$. In the pretest, the ERAM had omega values of $\omega_h = 0.25$ and $\omega_t = 0.75$; and in the follow-up of $\omega_h = 0.24$ and $\omega_t = 0.82$, suggesting acceptable reliability when considering the proportion of test variance of general ERA as well as the twelve emotions ($\omega_t$), and multidimensionality of the ERAM (see also *Flora, 2020*). The Pearson correlation coefficient (two-sided) between pretest and follow-up for the ERAM total scores was moderate to large, $r(79) = 0.58$, $p = 0.000$,

95% CI [0.40–0.72], suggesting the measurements are relatively similar. In two evaluation studies (*Laukka et al., 2021*), the ERAM showed acceptable to good reliability (study 1: $\alpha = 0.74$, $\omega_h = 0.69$, $\omega_t = 0.78$; study 2: $\alpha = 0.80$) and correlations with related concepts, suggesting criterion validity of the ERAM.

For assessing the accuracy for recognizing facial micro expressions, we used a computerized *micro expression recognition task* (MICRO; see also *Döllinger et al., 2021*) that consists of still pictures of facial emotional expressions that are shown for 200 ms and double-masked by neutral expressions (2s), taken from the *Radboud Faces Database* (*Langner et al., 2010*). For each measurement, 70 items are randomly chosen from a pool of 312 items and the participants have to decide which micro expression was depicted using a list of seven emotions: *happiness, surprise, fear, disgust, sadness, anger, contempt.* The task is based on Ekman's (see, *e.g.*, *Ekman & Cordaro, 2011*) theory of basic emotions and was conducted in Swedish. In the present sample, the internal consistencies were acceptable to good: $\alpha_{pre} = 0.78$ and $\alpha_{follow\ up} = 0.80$, respectively. In the pretest, the MICRO had omega values of $\omega_h = 0.26$ and $\omega_t = 0.82$; and $\omega_h = 0.29$ and $\omega_t = 0.87$ in the follow-up. Following a similar pattern as the ERAM, the $\omega_t$ values suggest good reliability when considering general micro expression ERA in combination with the seven emotions as factors and multidimensionality of the test. The correlation between pretest and follow-up measurement was of medium size, $r = 0.45$, $p = 0.00$, 95% CI [0.24–0.62], $n = 72$.

For the assessment of state affectivity, we used a Swedish translation of the *Positive and Negative Affect Schedule* (PANAS; *Watson, Clark & Tellegen, 1988*), a reliable, valid and often-used questionnaire for assessing positive and negative affectivity consisting of ten positive and ten negative emotional adjectives, and the task to indicate on a 5-point Likert scale (1 = *very slightly* to 5 = *very much*) to which extent each adjective describes the own emotional state. In this study, we found the positive subscale to have good internal consistency ($\alpha_{pos\_pre} = 0.83$; $\alpha_{pos\_follow\ up} = 0.87$) and the negative subscale to have acceptable to good internal consistency ($\alpha_{neg\_pre} = 0.75$; $\alpha_{neg\_follow\ up} = 0.83$).

## Data analysis

Data preparation and analysis were performed using R (*R Core Team, 2022*, v. 4.2.2) and RStudio (*RStudio Team, 2022*, v. 12.0+353). To test for sample differences between the trainee psychologists and the control group, independent *t*-tests and Wilcoxon-Mann-Whitney *U*-tests were conducted (Holm adjusted for multiple comparisons). We calculated *Wagner*'s *(1993)* unbiased hitrate ($H_u$) for the ERAM and MICRO scores to control for response bias (correcting for how often an emotion category was used incorrectly). Linear regression analyses were conducted to test for the influence of age, gender and state affectivity on ERA. Research shows that age and gender can influence ERA (see, *e.g.*, *Cortes et al., 2021*; *Thompson & Voyer, 2014*) and that even affective state can bias ERA, albeit the results in this field are somewhat contradictory (see, *e.g.*, *Manierka et al., 2021*; *Schmid & Mast, 2010*). For calculating internal consistency values of the measures, we used Kuder Richardson Formula 20 for dichotomous data (KR-20; *Kuder & Richardson, 1937*) for the ERA measures and Cronbach's alpha for the questionnaires. Omega total and $\omega_h$ were calculated according to *Revelle (2022*; see also *McNeish, 2018*).

Pearson's correlation coefficient was calculated for measuring the strength of association between pretest and follow-up. The alpha level for significance tests was set at 5%, however, for transparency we also report exact $p$-values.

To test *hypothesis 1*, independent $t$-tests and Wilcoxon-Mann-Whitney $U$-tests (Holm adjusted) were performed. To analyse possible reasons for dropout, a multiple logistic regression analysis was performed (continuous variables were centered). For investigating the changes in ERA from pretest to follow-up for the two groups (*hypothesis 2)*, we used mixed multilevel modeling (growth curve modeling) and step-wise model building (see, e.g., *Field & Wright, 2011*; *Finch, Bolin & Kelley, 2019*). Since we had considerable dropout, we used maximum likelihood estimation to handle missing data (*Enders, 2011*; *Enders, 2022*). However, as a sensitivity analysis, we also replicated the analysis without missing data methodology purely based on the sample that performed both pretest and follow-up (excluding the participants that dropped out completely). Because we only had two measurements and chose a fixed slope, we did not specify a variance-covariance structure. The assumptions for mixed multilevel modeling were fulfilled.

We started out modeling the unconditional means model including only a random intercept (allowing for ERA pretest values to vary between participants) and then compared it to two unconditional growth models including linear time (anchored at baseline: pretest = 0, follow-up = 1): an unconditional growth model with a random intercept and a fixed slope and then an unconditional growth model with a random intercept and a random slope (allowing for the individual ERA trajectories from pretest to follow-up to vary). For the full models (conditional growth models), we added *group* (trainee therapists = 1, control group = 0) as linear time-invariant predictor to the unconditional growth models with a fixed slope, since the fixed slope models had a better fit than the random slope models. For model comparison, we used Akaike's information criterion (AIC), since this goodness-of-fit estimate corrects for model complexity. To test for statistical differences between models, we used Analysis of Variance (ANOVA). The fixed effects *time* by *group* interaction indicates whether there is a difference in ERA change between the groups. For estimating a standardized effect sizes for the group differences in change scores, we used *Feingold*'s *(2009, 2013)* recommendation and subtracted the marginal contrast estimates from pretest to follow-up of the control group divided by the standard deviation at pretest from the marginal contrast estimates (pretest to follow-up) of the trainee psychotherapists divided by the standard deviation at pretest.

Even though dependent on the discipline and somewhat arbitrary, standardized effects sizes were interpreted according to *Cohen*'s *(1988)* suggestions: $d/d_z = 0.2$ (small), $d/d_z = 0.5$ (moderate) and $d/d_z = 0.8$ (large); $\eta^2/\eta^2_g = 0.01$ (small), $\eta^2/\eta^2_g = 0.06$ (moderate), $\eta^2/\eta^2_g = 0.14$ (large); $r = 0.10$ (small), $r = 0.30$ (moderate), $r = 0.50$ (large). In addition to $R$'s base packages, the following $R$ packages were used: *apa* (*Gromer, 2020*), *apaTables* (*Stanley, 2021*), *car* (*Fox & Weisberg, 2019*), *DescTools* (*Signorell et al., 2021*), *dplyr* (*Wickham et al., 2022*), *effects* (*Fox & Weisberg, 2019*), *emmeans* (*Lenth, 2023*), *ez* (*Lawrence, 2016*), *ggplot2* (*Wickham, 2016*), *ggpubr* (*Kassambara, 2020*), *Hmisc* (*Harrell, 2021*), *modelbased* (*Makowski et al., 2020*), *nlme* (*Pinheiro, Bates & R Core Team,*

*2022*), *psych* (*Revelle, 2022*), *rstatix* (*Kassambara, 2021*), *sjstats* (*Lüdecke, 2021*), *tidyr* (*Wickham & Girlich, 2022*), *tidyverse* (*Wickham et al., 2019*).

## RESULTS

Table 2 displays descriptive statistics and group comparisons for ERAM and MICRO, as well as for the ERAM modalities and valence and arousal categories, for the two time points. Since there were moderately sized significant age differences between trainee psychotherapists and the control group (Table 1), we conducted linear regression analyses, but did not find any influence of age on ERA at any time point for either test and not on the ERA change scores (see Supplemental Material). There were no group differences when it comes to state affectivity at any time point (see Table 1) and linear regression analyses did not indicate any influence of affectivity on ERA (see Supplemental Material). There were no significant gender differences for ERA at either test time point, and no influence of gender on ERA (see Supplemental Material). Further, there were no differences in ERA between the PDT and CBT students in the trainee psychotherapist group (see Supplemental Material). In the Supplemental Material, the reader also finds descriptive statistics for the additional questionnaires that the trainee psychotherapists filled in at pretest and how they correlate with ERA (Fig. S1).

### Hypothesis 1

The results of the group comparisons at pretest ($N$ = 154; Table 2) showed that there was a significant moderately sized difference between the trainee psychotherapists and the control group when it comes to detecting emotions in multiple modalities (ERAM total score). The trainee psychotherapists in the beginning of training were superior to the control, which also can be seen in the non-overlapping 95% confidence intervals (see Table 2 and Fig. 1A). When looking at the ERAM in detail (Table 2), we see that the trainee psychotherapists were significantly better at detecting nonverbal emotional expressions in the audio and the audio-video modality. They were also significantly better in both valence categories (positive as well as negative valence emotional expressions) and both arousal categories (high as well as low arousal emotional expressions). The groups showed equally good ERA only in the video modality. These results indicate that trainee psychotherapists are better at detecting emotional expressions in multiple modalities than the undergraduate students, especially when nonverbal audio content is involved. The same pattern can be observed when considering the 95% confidence intervals, which are non-overlapping for all variables but the video modality and positive valence.

The standardized effect sizes range from small to moderate and the 95% confidence intervals of the effect sizes based on bootstrap percentiles do not include zero, apart from the video modality (see Table 2). There was also a significant small difference when it comes to correctly identifying micro expressions (MICRO; see Table 2). Even here, the trainee psychotherapists in the beginning of their training showed significantly higher accuracy rates than the control group. It has to be noted, though, that the 95% confidence intervals overlap slightly (see Fig. 1B and Table 2). Still, the 95% confidence interval of the effect size does not include zero.

**Table 2** ERA test variables: descriptive statistics (means, standard deviations, 95% confidence intervals, sample size) and group comparisons (two-sided Student's *t*-test and two-sided Wilcoxon-Mann-Whitney *U*-test) of the observed ERA data.

| Measures | Trainee psychotherapists | Control group | Total | Statistic | Effect size |
|---|---|---|---|---|---|
| | *M (SD) [95% CI]* | *M (SD) [95% CI]* | *M (SD) [95% CI]* | *t\U [95% CI]* | *d/r [95% CI][b]* |
| **PRE ( N = 154)** | **n = 49** | **n = 105** | | | |
| ERAM total | 0.45 (0.08) [0.42–0.47] | 0.39 (0.09) [0.38–0.41] | 0.41 (0.09) [0.40–0.42] | *t(152) = 3.64* *p < 0.001\*\*\** [0.25–0.86] | *d = 0.63* [0.32–0.98] |
| ERAM audio | 0.43 (0.11) [0.40–0.46] | 0.36 (0.11) [0.34–0.38] | 0.38 (0.11) [0.36–0.40] | *t(152) = 3.84* *p < 0.001\*\*\** [0.04–0.11] | *d = 0.67* [0.34–1.03] |
| ERAM video | 0.42 (0.11) [0.39–0.46] | 0.42 (0.12) [0.40–0.45] | 0.42 (0.11) [0.41–0.44] | *t(152) = 0.15* *p = 0.88* [−0.04 to 0.4] | *d = 0.03* [−0.30 to 0.36] |
| ERAM audio-video | 0.64 (0.13) [0.61–0.68] | 0.55 (0.13) [0.53–0.57] | 0.58 (0.13) [0.56–0.60] | *t(152) = 4.26* *p < 0.001\*\*\** [0.05–0.14] | *d = 0.74* [0.40–1.08] |
| ERAM positive valence | 0.47 (0.11) [0.44–0.50] | 0.43 (0.12) [0.41–0.46] | 0.44 (0.12) [0.43–0.46] | *t(152) = 2.05* *p = 0.04\** [0.00–0.08] | *d = 0.35* [0.02–0.69] |
| ERAM negative valence | 0.43 (0.09) [0.40–0.46] | 0.36 (0.11) [0.34–0.39] | 0.39 (0.11) [0.37–0.40] | *t(152) = 3.58* *p < 0.001\*\*\** [0.03–0.10] | *d = 0.62* [0.31–1.01] |
| ERAM high arousal | 0.46 (0.10) [0.43–0.49] | 0.41 (0.11) [0.39–0.43] | 0.43 (0.11) [0.41–0.44] | *U = 3,182* *p = 0.02\** [0.01–0.08] | *r = 0.19* [0.04–0.33] |
| ERAM low arousal | 0.64 (0.09) [0.61–0.66] | 0.58 (0.11) [0.56–0.60] | 0.60 (0.11) [0.58–0.62] | *U = 3,358* *p = 0.002\*\** [0.03–0.08] | *r = 0.25* [0.09–0.4] |
| MICRO total | 0.50 (0.14) [0.46–0.54] | 0.45 (0.15) [0.42–0.54] | 0.46 (0.15) [0.44–0.49] | *t(152) = 2.02* *p = 0.05\** [0.00–0.10] | *d = 0.35* [0.03–0.76] |
| **FOLLOW-UP ( n = 72)** | **n = 31** | **n = 41** | | | |
| ERAM total | 0.45 (0.09) [0.42–0.47] | 0.45 (0.10) [0.43–0.47] | 0.45 (0.09) [0.43–0.47] | *t(70) = 0.19* *p = 0.85* [−0.05 to 0.04] | *d = −0.04* [−0.48 to 0.44] |
| ERAM audio | 0.41 (0.10) [0.39–0.44] | 0.39 (0.11) [0.39–0.44] | 0.40 (0.10) [0.37–0.42] | *t(70) = 1.05* *p = 0.30* [−0.02 to 0.08] | *d = 0.25* [−0.24 to 0.75] |
| ERAM video | 0.43 (0.13) [0.39–0.47] | 0.49 (0.13) [0.46–0.51] | 0.46 (0.14) [0.43–0.49] | *t(70) = −1.74* *p = 0.09* [−0.12 to 0.01] | *d = −0.41* [−0.92 to 0.02] |
| ERAM audio-video | 0.67 (0.13) [0.63–0.70] | 0.63 (0.14) [0.60–0.70] | 0.64 (0.14) [0.61–0.68] | *U = 707* *p = 0.42* [−0.04 to 0.09] | *r = 0.10* [0.00–0.34] |
| ERAM positive valence | 0.47 (0.15) [0.43–0.51] | 0.50 (0.12) [0.47–0.52] | 0.48 (0.13) [0.45–0.51] | *t(70) = −0.94* *p = 0.35* [−0.09 to 0.03] | *d = −0.22* [−0.72 to 0.3] |

(Continued)

| Table 2 (continued) | | | | | |
|---|---|---|---|---|---|
| Measures | Trainee psychotherapists | Control group | Total | *Statistic* | *Effect size* |
| | *M (SD)* [95% CI] | *M (SD)* [95% CI] | *M (SD)* [95% CI] | *t\U* [95% CI] | *d/r* [95% CI][b] |
| PRE ( *N* = 154) | *n* = 49 | *n* = 105 | | | |
| ERAM negative valence | 0.44 (0.08) [0.41–0.46] | 0.42 (0.10) [0.40–0.44] | 0.43 (0.09) [0.41–0.45] | $t(70) = 0.66$ $p = 0.51$ [−0.03 to 0.06] | $d = 0.16$ [−0.33 to 0.64] |
| ERAM high arousal | 0.46 (0.11) [0.43–0.49] | 0.45 (0.11) [0.43–0.47] | 0.45 (0.11) [0.43–0.48] | $U = 661$ $p = 0.78$ [−0.05 to 0.06] | $r = 0.03$ [0.00–0.28] |
| ERAM low arousal | 0.64 (0.11) [0.61–0.67] | 0.66 (0.11) [0.63–0.67] | 0.65 (0.11) [0.63–0.67] | $U = 566$ $p = 0.43$ [−0.08 to 0.03] | $r = 0.09$ [0.01–0.34] |
| MICRO total | 0.54 (0.16) [0.49–0.58] | 0.60 (0.16) [0.57–0.63] | 0.58 (0.16) [0.54–0.61] | $U = 458$ $p = 0.04^*$ [−0.14 to −0.00] | $r = 0.24$ [0.03–0.45] |

Note:
For the mixed multilevel modeling analyses, missing data were handled *via* maximum likelihood estimation. Table 2 presents the observed scores. [95% CI][b]: 95% Confidence Interval for effect size is based on 1,000 bootstrap resamples of the mean difference (percentile interval). Common standardized effect size estimates: Cohen's d: $d = 0.2$ (small effect), $d = 0.5$ (moderate effect) and $d = 0.8$ (large effect); Wilcoxon-Mann-Whitney U-test: $r = 0.10$ (small), $r = 0.30$ (moderate), $r = 0.50$ (large).
$^*p < 0.05$, $^{**}p < 0.01$, $^{***}p < 0.001$ (Holm adjusted).

## Exploratory analysis of dropout rate

Eighty-two participants did not participate in the follow-up measurement about one and a half years after the pretest. Sixty-one percent of the control group and 37% of the trainee psychotherapists had dropped out. The higher dropout rate for the control group was expected, as the undergraduate students often only read a few courses at the psychology department and therefore are harder to convince to come back for follow-up measurements, unlike the trainee psychotherapists. Beyond that, we put several possible reasons for dropout to the test, namely that those participants who realized that they were not very good at recognizing nonverbal emotional expressions in others or those that were in a negative mood during pretest did not feel motivated to come back to the lab a second time and that this evened out differences between the two groups. Also, age could play a role. To explore whether dropout was connected to low ERAM or MICRO baseline values, positive or negative affectivity at pretest, group affiliation or age, we conducted a multiple logistic regression analysis. First, the full model (group affiliation, age, ERAM total at pretest, MICRO at pretest, PANAS positive at pretest, PANAS negative at pretest) was compared to the null model. The full model was significantly different from the null model, $\chi^2(147, N = 154) = 22.31$, $p < 0.001$, and had a better fit (full model: AIC = 206.5, BIC = 230, Nagelkerke's $r^2 = 0.18$; null model: AIC = 216.8, BIC = 222.9, Nagelkerke's $r^2 = 0.00$). However, in the full model, only age turned out to be a significant predictor ($p < 0.001$). The likelihood of continuing with the follow-up measurement increased slightly with age (OR = 1.07, 95% CI [1.02–1.13]), or, in other words, the likelihood of dropout increased slightly the younger participants were. Since only age was significant, we also tested a simple logistic regression model with age as predictor, but it was not statistically different

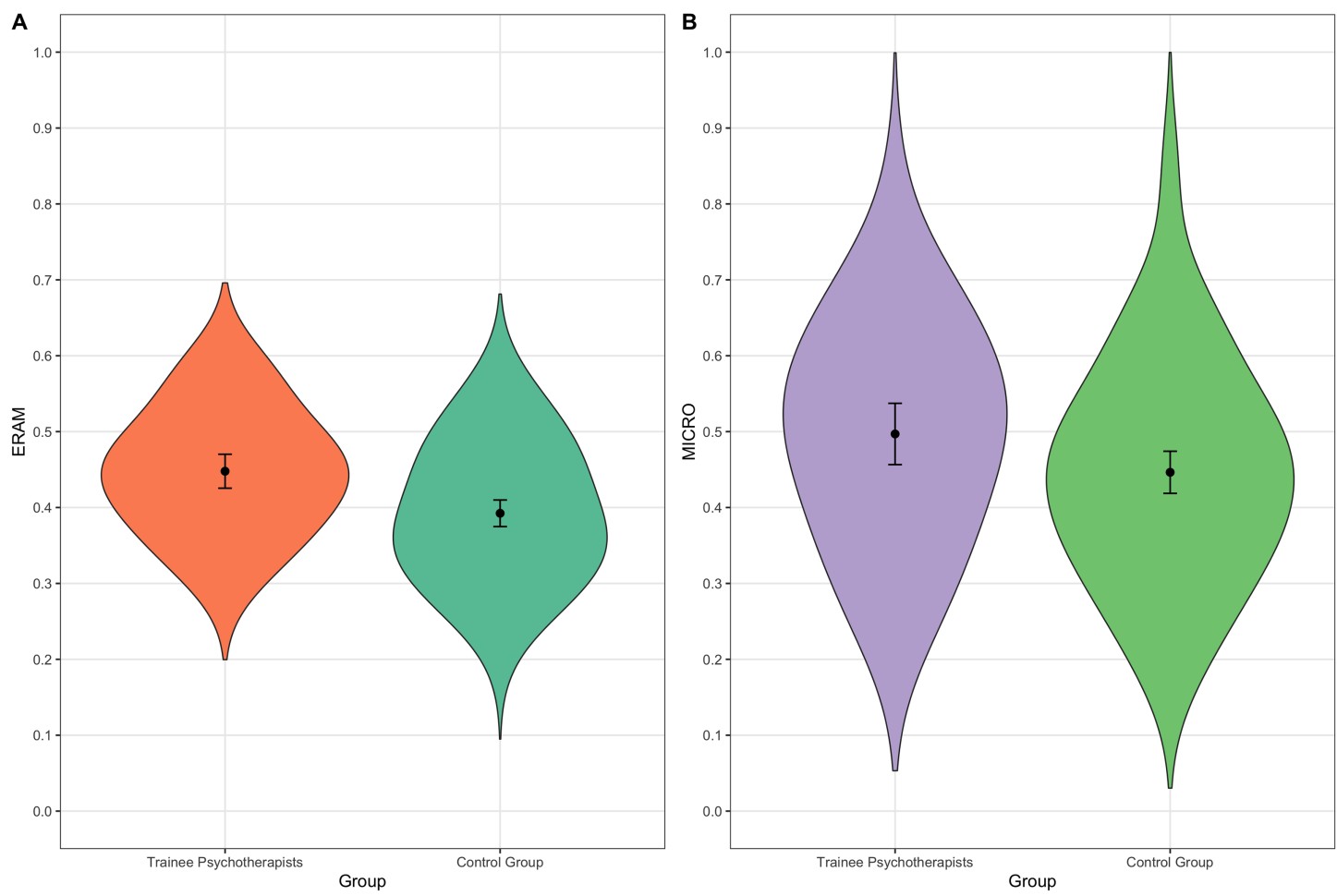

**Figure 1 Violin plots of ERA per group at pretest.** (A) ERAM pretest. $N = 154$. (B) MICRO pretest. $N = 154$. Note. The violin plots display the kernel probability density of the data at pretest with means (points) and 95% confidence intervals.

from the full model ($\chi^2(152, N = 154) = 199.78$, $p < 0.10$; age-only model: AIC = 205.8, BIC = 214.9, Nagelkerke's $r^2 = 0.11$; OR = 1.09, 95% CI [1.04–1.14]).

## Hypothesis 2

Table 2 shows descriptive statistics of the observed data and group comparisons at follow-up ($n = 72$). Figure S2 shows the individual ERA trajectories for all participants (observed data). To test whether the trainee psychotherapists improved their ERA during the practical clinical training more than the control group (*hypothesis 2*), we conducted two mixed multilevel modeling analyses with maximum likelihood estimation (see *Data analysis*), one for the ERAM total score and one for the MICRO. The estimated marginal means at follow-up were very similar to the observed data. The trainee psychotherapists had values of $EMM = 0.45$, $SE = 0.01$, 95% CI [0.42–0.48] for the ERAM and $EMM = 0.56$, $SE = 0.03$, 95% CI [0.51–0.61] for the MICRO. The control group had values of $EMM = 0.44$, $SE = 0.01$, 95% CI [0.41–0.46] for the ERAM and $EMM = 0.60$, $SE = 0.02$, 95% CI [0.56–0.64] for the MICRO. There were no significant between-groups differences

**Table 3 Fixed effects: linear mixed-effects model fit by maximum likelihood estimation.**

|  | Value | SE | Df | t-value | p-value | 95% CI |
|---|---|---|---|---|---|---|
| **ERAM (total)** | | | | | | |
| Intercept | 0.39 | 0.01 | 152 | 44.92 | 0.00*** | [0.38–0.41] |
| Time | 0.05 | 0.01 | 70 | 3.88 | 0.00*** | [0.02–0.07] |
| Group | 0.06 | 0.02 | 152 | 3.58 | 0.00*** | [0.03–0.09] |
| Time * group | −0.04 | 0.02 | 70 | −2.37 | 0.02* | [−0.08 to −0.01] |
| **MICRO** | | | | | | |
| Intercept | 0.45 | 0.01 | 152 | 30.72 | 0.00*** | [0.42–0.47] |
| Time | 0.15 | 0.02 | 70 | 6.87 | 0.00*** | [0.11–0.20] |
| Group | 0.05 | 0.03 | 152 | 1.96 | 0.05* | [0.00–0.10] |
| Time * group | −0.09 | 0.03 | 70 | −2.54 | 0.01** | [−0.16 to −0.02] |

**Note:**
Number of Observations: 226; Number of Groups: 154. $^*p < 0.05$, $^{**}p < 0.01$, $^{***}p < 0.001$.

at follow-up after maximum likelihood estimation, $t(70) = -0.63$, $p = 0.99$ for the ERAM and $t(70) = 1.11$, $p = 0.27$ for the MICRO, and the 95% confidence intervals overlapped.

When comparing different models for the ERAM, the model with the best fit was the full model (conditional growth model) with a random intercept, fixed slope and group as time-invariant predictor (AIC = −473.04), in comparison to the unconditional means model (AIC = −454.71), the unconditional growth model including a random slope (AIC = −458.98) and the unconditional growth model with a fixed slope (AIC = −462.94). There were significant ($p < 0.00$) statistical differences between the full model and the other models and the data was 14 times more likely under the full model than the next best model according to a Likelihood Ratio Test, $\chi^2(2) = 14.09$, $p = 0.00$). Similarly, in case of the MICRO, the model with the best fit was the full model (conditional growth model) with a random intercept, fixed slope and group as time-invariant predictor (AIC = −288.21), in comparison to the unconditional means model (AIC = −188.77), the unconditional growth model including a random slope (AIC = −221.73) and the unconditional growth model with a fixed slope (AIC = −224.72). There were significant ($p < 0.00$) statistical differences between the full model and the other models and the data was seven times more likely under the full model than the next best model, $\chi^2(2) = 7.50$, $p = 0.02$).

For the ERAM, there was a significant *group* by *time* interaction effect, suggesting that, depending on whether the participant belongs to the trainee psychotherapist group or the control group, the multimodal ERA trajectories are different (between-group difference in slope = −0.04, SE = 0.02, $t(70) = -2.37$, $p = 0.02$; 95% CI [−0.08 to −0.01]; see Table 3). The control group was estimated to significantly improve in multimodal ERA from pretest to follow-up with five percent points, $t(70) = -3.88$, $p < 0.001$, 95% CI [−0.07 to −0.02], whereas the trainee psychotherapists' slope remained flat, $t(70) = -0.18$, $p = 0.85$, 95% CI [−0.03 to 0.03] (Holm adjusted $p$-values). See Fig. 2A for a visual display of the multimodal ERA slopes of the two groups. The difference in change scores can be interpreted as moderate ($d = 0.63$). Even for the MICRO, there was a significant *group* by

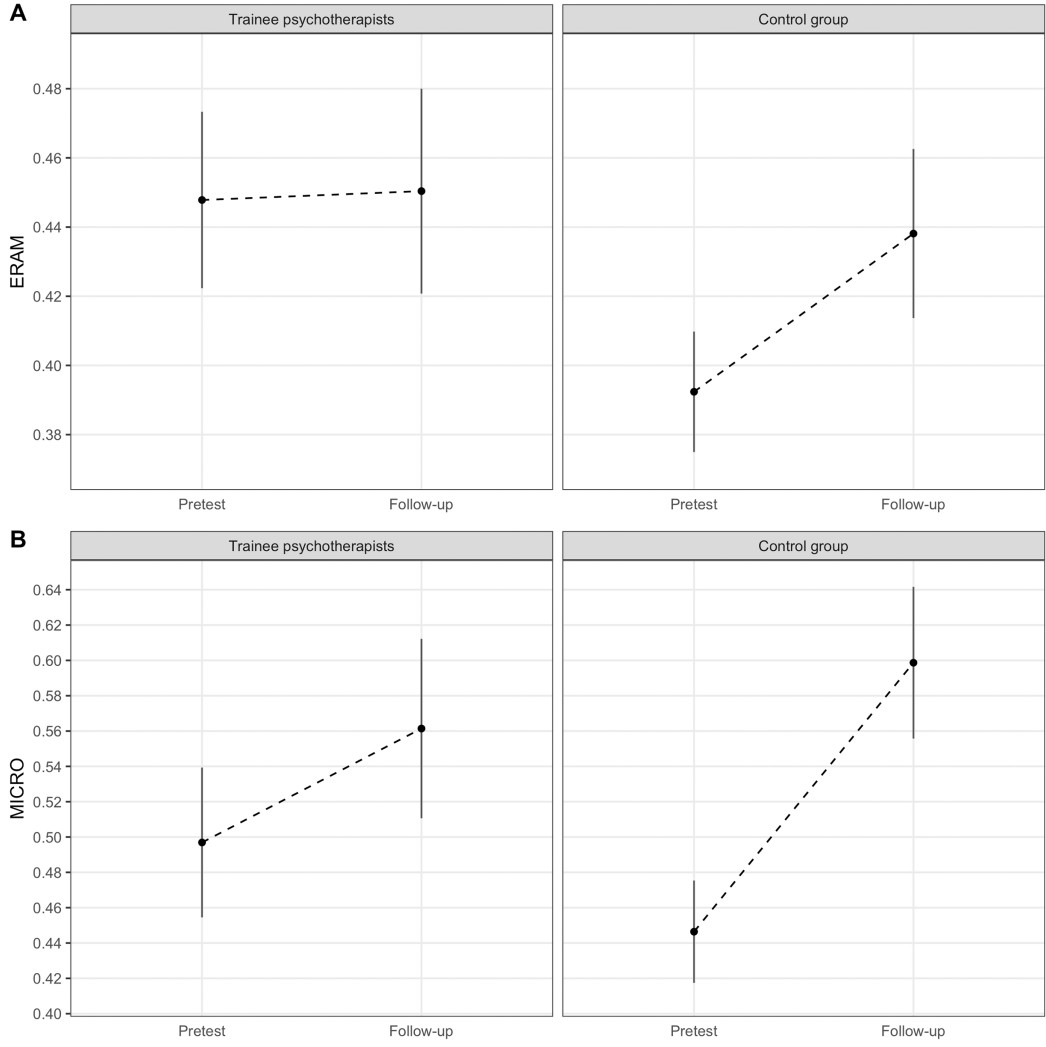

**Figure 2  Line graphs of the ERA pretest and follow-up data.** (A) ERAM change from pretest to follow-up. (B) MICRO change from pretest to follow-up. Note. Based on estimated marginal means. Error bars represent 95% Confidence Intervals.

*time* interaction, (between-group difference in slope = −0.09, $SE$ = 0.03, $t(70)$ = −2.54, $p$ = 0.01; 95% CI [−0.16 to −0.02]; see Table 3). The within-group differences from pretest to follow-up were both significant. The trainee psychotherapists estimated improvement in micro expression ERA was 6%, $t(70)$ = −2.43, $p$ = 0.02, 95% CI [−0.12 to −0.01] and the control groups' improvement was estimated to be 15%, $t(70)$ = −6.87, $p$ < 0.001, 95% CI [−0.20 to −0.11]. Figure 2B shows that both groups improve from pretest to follow-up but that the upwards slope of the control group is steeper. The standardized effect size for the difference in change trajectories can be interpreted as moderate ($d$ = 0.57).

In the Supplemental Material, the reader finds replications of the two mixed multilevel modeling analyses after excluding the participants that dropped out (without missing data handling method). The sensitivity analyses confirm the initial results that the trainee psychotherapists do not improve in multimodal or micro expression ERA more than the control group, and, thus, show that the results are robust. After excluding the dropouts, the

differences between the trainee psychotherapists and the control group were even less pronounced. The reader also finds exploratory additional mixed multilevel models. One in which we added age as predictor and one in which we divided the trainee psychotherapists into PDT and CBT students before comparing those two groups to the control group. The full models reported above including only group (trainee psychotherapists *vs* control group) as predictor showed the best model fit. Age was not found to be a significant predictor of the ERA slopes.

## DISCUSSION

In this study, we investigated trainee psychotherapists' ERA in the beginning of practical clinical training and at the end of it in comparison to a group of undergraduate students. In the beginning of practical clinical training, the trainee psychotherapists indeed showed superior ERA to the control group. They were significantly better at detecting emotions in multiple modalities as well as micro expressions. The standardized effect sizes were moderate (ERAM total) and small (MICRO), respectively. Even when investigating the results for the ERAM modalities and valence and arousal categories, the group differences were very consistent. The results indicate that trainee psychotherapists are better at detecting positive as well as negative valence items and high as well as low arousal items. Further, they suggest that trainee psychotherapists are principally better at detecting emotions when (also) nonverbal audio information is provided (audio modality and audio-video modality), but that there is no difference when only video information is used, leading us to the cautious suggestion that auditory nonverbal information seems to be particularly useful for this group. However, because of the lower item numbers when investigating the ERAM modalities separately, this finding should not be overstated. Our findings are in line with *hypothesis 1* (trainee psychotherapists' ERA at pretest > control group's ERA).

Since the trainee psychotherapists were just in the beginning of their clinical training, we interpret these differences as general differences in ERA between the groups. Even though there were significant age differences between the groups, age did not influence ERA at any time point or the ERA change scores. Thus, we do not believe that the differences in ERA are a consequence of the trainee psychotherapists being slightly older than the control group. Future research will need to determine why trainee psychotherapists show higher ERA. Possible reasons for their superior ERA could, *e.g.*, be certain traits associated with the profession or choice of profession, motivational aspects, heightened interest for or attention towards other people's emotions (maybe the trainee psychotherapists had a "headstart" because they were already interested in other people's emotions prior to studying clinical psychology), educational variables related to clinical psychology and psychotherapy (*e.g.*, prior theoretical knowledge about psychotherapy even before choosing an approach and starting to practice it), or due to other educational variables apart from specified (PDT or CBT) psychotherapy education. Future studies should also try to investigate and maybe control for certain variables that we think could have influenced the results for *hypothesis 1*, *e.g.*, duration of higher education (the trainee psychotherapists were in the later stage of their education whereas the control group

generally in an earlier stage), exposure to similar socio-emotional lab tests and possible learning effects (maybe the control group had participated in fewer research projects than the trainee psychotherapists due to different stages of education), and personality variables and other (interpersonal) characteristics that could be related to ERA.

Due to the mixed previous findings, we also need to consider methodological reasons for finding trainee psychotherapists in the beginning of their education to be superior in ERA. Our findings are in line with *Pauza et al.*'s *(2010)* study that found that psychotherapy trainees in the beginning of their education were better at detecting brief (300 ms) facial emotional expressions than several control populations (incl. a normal population sample and trainees of other professions). Like the MICRO, the test that *Pauza et al. (2010)* used was a computerized task including basic emotions, but the emotions were shown for a longer duration and were only single-masked. Further, the study did explicitly investigate trainee psychotherapists of several therapeutic orientations, which makes it relatively comparable with the present study. However, also *Hutchison & Gerstein (2012)* and *Hassenstab et al. (2007)* used pictures of facial expressions of basic emotions by Ekman's group, but these studies did not find a difference between trainee psychotherapists and undergraduate students (*Hutchison & Gerstein, 2012*), or between practicing psychotherapists and matched controls (*Hassenstab et al., 2007*), respectively. Tendentially, *Pauza et al.*'s *(2010)* study and the present study had bigger sample sizes, which might have influenced the power to find significant effects. Our results are also in line with *Machado, Beutler & Greenberg (1999)* study that found a difference in nonverbal ERA between experienced psychotherapists and undergraduate psychology students. *Machado, Beutler & Greenberg*'s *(1999)* methodology differed from that of the present study's; they asked their participants to rate segments of a video-recorded therapy session instead of using a standardized computerized ERA task. In our opinion, the fact that different methodologies lead to similar results strengthens the trustworthiness of the finding that (trainee) psychotherapists possess higher ERA than undergraduate student populations, even if the reasons for this are still not clear.

*Hypothesis 2* considered whether trainee psychotherapists' ERA increased more from pre- to follow-up measurement than the control groups', theoretically due to one and a half years of psychotherapy training. The results of two mixed multilevel modeling analyses using maximum likelihood estimation suggest that this was not the case. There was a significant interaction effect between *group* and *time* for the ERAM, indicating between-group differences in multimodal ERA change trajectories. However, this effect was not in the hypothesized direction. Contrary to expectation, the control group was estimated to improve significantly from pretest to follow-up (5%), while the trainee psychotherapists would not change at all. The difference in change slopes was of moderate size. The control group improved their ERA in so far that they now were on the same multimodal ERA level as the trainee psychotherapists. There was also a significant interaction effect for the MICRO suggesting that the micro expression ERA slopes differed. Both groups were estimated to significantly improve their ability to detect micro expressions in others from pretest to follow-up, but the control group's improvement was estimated to be steeper. The trainee psychotherapists' estimated increase was 6% whereas

the control group's was 15%. The magnitude of change between the two groups varied significantly and was of moderate size. Thus, also for micro expression ERA, the hypothesis that trainee psychotherapists improve their ERA more strongly than the control group should be rejected.

Since the trainee psychotherapists did not improve in ERA during the one and a half years of psychotherapy education, in case of the multimodal ERA not at all and in case of the micro expression ERA not to a greater degree than the control group, we need to discuss possible reasons for that. Psychotherapy education at Stockholm university does not include any explicit ERA training in form of standardized training procedures. When socio-emotional or interpersonal abilities or skills are not explicitly trained, it might be unsurprising that no improvement took place. This might be particularly true for ERA facets, like multimodal ERA and micro expression ERA, that are assessed *via* standardized emotion perception and interpretation tasks. Both psychotherapy approaches at Stockholm University, PDT and CBT, focus on patients' emotions in supervision, in the theoretical education and in the therapies themselves. The practical and theoretical education includes discussions about patients' emotions, *e.g.*, in case formulations, treatment plans and weekly video-assisted supervision, and practical exercises or role plays. All this had led us to hypothesize that some degree of improvement would still take place. However, it is possible that the ERAM and the MICRO are not capable of capturing specific components of ERA that are relevant in the clinical context or in naturalistic interpersonal situations in general. This could also explain why our findings deviate from *Machado, Beutler & Greenberg*'s *(1999)* finding that clinical experience was associated with higher ERA. *Machado, Beutler & Greenberg*'s *(1999)* used a more naturalistic procedure to measure ERA in the psychotherapeutic setting. Possibly, we were not able to find those effects using standardized computerized ERA tasks. Thus, our findings do not necessarily mean that the trainee psychotherapists did not improve in ERA in naturalistic therapy situations or improve their clinical skills of working with patients' emotions due to the education. On the other hand, it might also be plausible that the psychotherapy education lays too much emphasis on talking about verbal expressions of emotion instead of nonverbal displays, like body language, facial expressions and tone of voice, as captured by the ERAM and the MICRO, and that the trainee psychotherapists indeed did not improve in those important skills. It needs to be investigated whether other, more affect-focused forms of psychotherapy, like *Intensive Short-Term Psychodynamic Psychotherapy* (*Davanloo, 2001*), *Accelerated Experiential-Dynamic Psychotherapy* (*Fosha & Yeung, 2006*) or *Emotion-focused Therapy* (*Greenberg, 2015*), that very explicitly attend to moment-to-moment emotional changes in session and explicitly train perceiving and working with those shifts in their educational programs (*e.g.*, in form of deliberate practice; see *Goldman, Vaz & Rousmaniere, 2021*), might lead to improvements in ERA that the PDT and CBT educations at Stockholm University did not.

Regarding multimodal ERA, it is peculiar that the trainee psychotherapists did not improve at all from pretest to follow-up, whereas the control group did. We considered the possibility of the trainees' stagnation being a ceiling effect, but do not deem this to be likely, as the data was roughly normally distributed (with about 50% showing more than average

ERA; see also *Laukka et al., 2021* for reference ERAM data). Generally, it is possible that the psychological development that occurs in young adulthood might have influenced the differences in multimodal ERA improvement, as the control group was a bit younger on average. Even though ERA was not influenced by age differences in this study, we cannot preclude that other age-related factor might have mediated the ERA change differences. Further, factors related to academic education, knowledge about psychological theory and participation in studies could have contributed to the control group's improvement. Even though they did not receive any theoretical or practical psychotherapy training, the exposure to and reflection about psychological topics, as well as possible participation in research projects for course credits, could have led to increased interest for and knowledge about emotions, emotional expressions and interpersonal topics, which could have influenced their ERA. This seems plausible since the undergraduate students advanced in multimodal ERA up to the same level as the trainee psychotherapists and because the trainee psychotherapists generally were in the later stages of their education compared to the undergraduate students (meaning that the control group could have "read up" on those topics a little later). Another possible interpretation is self-selection. The dropout in the control group was higher than in the trainee psychotherapist group, which could have something to do with the fact that the trainee psychotherapists have a stronger bond to the department (since they are studying at the same institution for 5 years) and might be more motivated to participate again. In addition, it might have been easier for them to come back for the follow-up for practical reasons. The participants in the control group that came back might have been those that were very motivated or those that continued their education at the department of psychology. They might have been especially gifted or eager students or students that enjoyed taking those kinds of tests, which may have led to better results. However, this reasoning could also be (partly) true for the trainee psychotherapists and we tried to take into account the missingness of data in our statistical analyses using maximum likelihood estimation of the missing values.

For the recognition of micro expressions, both groups were estimated to become significantly better, but the control group's improvement was estimated to be greater. It is unclear why the undergraduate students' improvement in micro expression ERA surpassed the trainee psychotherapists'. Theoretically, it is possible, that continued education, participation in research and reasons related to dropout might be applicable in the case of micro expression ERA as well. Still, the trainee psychotherapists did at least significantly improve in this ERA facet, even if not more than the control group. For both groups, drawing attention towards the existence of very brief facial expressions might have led to more interest in those more subtle emotional expressions, which could have contributed to a general improvement in this ability. Micro expressions, in contrast to multimodal emotional expressions, might be more of a novelty, which might also explain the discrepancy in the trainee psychotherapists' ERA improvements between the ERAM and MICRO. Even for micro expressions, the general improvement, as well as the differences in the slopes, could potentially also be a result of psychological maturation and improved socio-emotional skills during young adulthood. In general, it can be seen as positive that trainee therapists improve their micro expression ERA with time, even if this

result is not specifically related to the practical psychotherapy education. Micro expression detection, potentially, is an important socio-emotional skill in psychotherapy, as it could lead to more effective interventions (*e.g.*, exploration of unconscious aspects of emotions) and more client safety (*e.g.*, more information about patient's mental states and safety risks).

## Implications

We found no indication for the idea that basic psychotherapy education in PDT and CBT (at Stockholm university) contributes to improvements in trainee psychotherapists' ERA, neither for multimodal ERA nor for micro expression ERA. This has several possible implications. The first, as mentioned above, is the possible need for explicit training of ERA in the psychotherapy education. If ERA, as measured by standardized computerized tests, is related to psychotherapy outcome, and there is first indication to believe so (see *Abargil & Tishby, 2021*), even though more studies are needed to establish this with certainty, then it might be beneficial for psychotherapists and patients to incorporate ERA training in the psychotherapy education. This could be done, for example, using standardized computerized ERA training procedures (see *Döllinger et al., 2021*), group seminars about ERA using recordings of emotional expressions, or deliberate practice exercises (see *Rousmaniere, 2017*). Another possible implication is for trainee selection. In Sweden, as in many other countries, acceptance into the clinical psychology program is determined by the applicant's grades and other academic merits. If we assume that ERA does not improve during this type of practical psychotherapy education, as suggested by our results, and that it is an important therapist skill that is likely positively related to psychotherapy outcome (*Abargil & Tishby, 2021*), then (ecologically valid) ERA tests could potentially become part of assessment procedures for clinical psychology and psychotherapy programs, besides other important personal and professional capacities of effective therapists, like empathy or the capacity to repair alliances (see, *e.g.*, *Heinonen & Nissen-Lie, 2020*; *Wampold, 2015*) and other indicators.

## Strengths, limitations and future directions

The present study has several strengths and limitations. In comparison with the existing literature, the sample size for the first research question, was rather large and the use of a long-term follow-up measurement and a within-between design is generally positive. However, the group sizes were unbalanced and the longitudinal data collection led to considerable dropout, and, with that, to some uncertainty regarding the results and implications of *hypothesis 2*. The problem of attrition is an obvious limitation of the study. From pretest to follow-up, there was considerable dropout (53%) and the dropout rates between trainee psychotherapists and control group diverged. Even if high and uneven dropout rates were expected after about one and a half years, it makes it harder to draw valid conclusions from our study's results. We tried to statistically explore possible reasons for dropout and performed a multiple logistic regression analysis in which we found that participants' (young) age predicted dropout, even if only slightly. It has to be noted, though, that this was an exploratory additional analysis and that it cannot be precluded

that other factors might have influenced dropout. In future studies, possible influences for dropout should be assessed and, if possible, prevented. Nonetheless, we tried to address the problem of missing data by using a mixed multilevel modeling approach that handled missing data by estimating it with maximum likelihood estimation. Even if we cannot completely rule out that there were factors influencing dropout (data was likely not missing *completely* at random), we suppose that the missingness was not directly related to the outcome variable itself (*i.e.*, data was missing at random), which allows for this missing data estimation approach. Using a maximum likelihood estimation or multiple imputation is oftentimes preferable to deleting whole cases, which can potentially lead to bias (*Enders, 2011*). Still, because the dropout was very large and as a way to double-check our results (sensitivity analyses), we also performed mixed multilevel modeling after excluding the dropouts (without missing data handling method). These analyses confirmed our findings. Further, we added age as an additional predictor variable in an exploratory mixed multilevel model, but did not find age to be a significant predictor of the outcome.

We used two standardized, computerized ERA measures for different but related ERA facets (multimodal ERA and micro expression ERA), which provides a more detailed view on the subject and which gives our finding that psychotherapy education does not lead to enhanced ERA compared to a control group more validity. Since there is task-specific variance between ERA tests, it is recommended to administer several measures of ERA in one study and to compare their results (see, *e.g.*, *Olderbak et al., 2021*). Performance-based ERA tests (like the ERAM and MICRO) should be preferred to self-reports that are based on the participants' own perception of their ERA, which could potentially be influenced by biases. Research about emotional intelligence, for example, found no or only weak correlations between self-report and performance-based measures (*Goldenberg, Matheson & Mantler, 2006*). Especially the ERAM can be considered quite ecologically valid, as it uses dynamic stimuli and both visual and auditory items. It also provides the opportunity to compare ERA unimodally (visual or auditory items presented separately) to the same ERA modalities presented in a multimodal fashion (visual and auditory items presented together), which is rare in ERA research. In future studies it would be interesting to replicate this investigation with psychotherapists that use unimodal channels of communication with their patients, *e.g.*, in classical psychoanalysis with a lying setting (predominantly audio modality), tele-therapy (audio modality) and internet-delivered treatments (possibly limited body language). On a side note, in the aftermath of a pandemic, the need for an even more detailed view on certain aspects of single modalities becomes evident, like the investigation of psychotherapists' ERA for patients wearing face masks that cover the lower part of the face and only leave the eyes and the rest of the upper face as sources of information. Indeed, research suggests that psychotherapists' ERA for some but not all emotions is impaired for faces wearing face masks (*Bani et al., 2023*; see also *Mitzkovitz et al., 2022*).

For all that, we still have to note that the ERAM did not show good internal consistency according to the Cronbach's alpha values in this sample, which limits the interpretability of the findings regarding ERA in multiple modalities. Even if Cronbach's alpha was acceptable to good in previous evaluation studies (*Laukka et al., 2021*), this could indicate

that the ERAM shows some degree of instability. Yet, we argue that Cronbach's alpha alone possibly is not the most convincing estimate of reliability for performance-based ERA tasks including a high number of emotion subscales and items with varying intensity and difficulty levels. For that reason, we also presented omega estimates, which suggest that the ERAM is still a useful measure for ERA as well as its twelve emotion subscales (see Material). The pretest-follow-up correlation was moderate to large, suggesting that items that are recognized well at the first occasion are also recognized well at the second occasion. Regarding the MICRO, it needs to be noted that micro expression research, especially in the context of lie or deception detection, is not without criticism (see, *e.g.*, *Vrij, Hartwig & Granhag, 2019*; *Weinberger, 2010*). However, in the present study, we did not investigate micro expressions exclusively as signs of deception, but understand them as a broader emotional phenomenon that can occur in various contexts, *e.g.*, as sign of more or less conscious deception, as marker of defense against uncomfortable emotions, as a form of self-regulation, or other possible occurrences.

Further, it needs to be noted that neither the ERAM nor the MICRO items were tailored to psychotherapy contexts and that they included only posed emotional expressions (by actors or instructed laypeople, respectively). Even though well-produced portrayals of emotional expressions are useful and valid in emotion research (*Scherer & Bänziger, 2010*), they generally do not allow for more complex answering options, like allowing for the display and recognition of mixed emotions. In the current context, it would also be relevant to assess ERA in clinical situations, for example using real life therapy sessions (see, *e.g.*, *Machado, Beutler & Greenberg, 1999*) or ERA tasks that display clinical situations (see, *e.g.*, *Blanch-Hartigan, 2011*), to generate even more ecologically valid findings.

In this study, we included both PDT and CBT students. Still, these two therapy approaches are based on very different assumptions and interventions. Accordingly, the psychotherapy education for PDT and CBT can be quite different. Since there were no significant ERA differences between the PDT and CBT students, since we believe ERA to be a factor important for all kinds of therapy approaches, and as to not lose statistical power, we did not investigate those two groups separately. Thus, our results are only interpretable in a general psychotherapy education framework, even though it might also be interesting to investigate those two groups separately (see Supplemental Material for exploratory analyses). Further, we cannot generalise findings about trainee psychotherapists to all other therapist populations, like more experienced therapists or therapists that are working with specialized therapeutic approaches. However, beyond practicality issues, the investigation of trainee psychotherapists' ERA is interesting in itself, as we know from meta-analyses that the correlation between therapist empathy and psychotherapy outcome is larger for less experienced psychotherapists (see, *e.g.*, *Elliott et al., 2011*), leading to the suggestion that psychotherapists with less experience and knowledge about psychotherapy (and techniques) rely more strongly on factors within themselves, like ERA. Other limitations to the generalizability of the present results include the specific characteristics of the psychotherapy education at Stockholm university. Psychotherapy education programs vary in their duration, teaching and learning methods, assessment procedures, and other characteristics. Thus, we cannot infer that

psychotherapy education in general does not or cannot lead to ERA improvements. It is, *e. g.*, possible, that longer or more intensive programs or more supervised therapy experience could lead to changes in ERA. Finally, it is relevant to find out more about the process of learning certain psychotherapeutic skills and about therapist factors that might (directly or indirectly) influence the clinical interaction. Research should not only investigate which therapist factors are of relevance for adequate psychotherapeutic treatments, but also examine how effective psychotherapy education actually is and how clinical training could become more effective and efficient in teaching favourable therapist abilities and skills.

## CONCLUSIONS

In conclusion, from psychotherapy outcome and process research, we know that working with emotions in psychotherapy is an important aspect of psychotherapeutic work in various therapeutic approaches. The hypotheses of the present study were that trainee psychotherapists show superior ERA to a control group of undergraduate students at the pretest and that their ERA would increase more strongly than the control groups' during the 1.5 years of psychotherapy education. Our findings suggest that trainee psychotherapists indeed possess higher ERA than undergraduate students at baseline. However, we could not find support for the idea that the trainee psychotherapists' ERA would improve more strongly than the control groups' ERA. This suggests that specified theoretical and practical psychotherapy education and clinical do not necessarily lead to further improvements in ERA for the trainee psychotherapists, at least not as conducted in the present study. We conclude that future research should investigate more explicit and systematic ways to train (trainee) psychotherapists' ERA, *e.g.*, *via* standardized computerized trainings, clinical group exercises or deliberate practice. There is also the need for replication of these results with improved research designs that lead to lower dropout rates and that use more reliable or additional measures of ERA, possibly tailored to the psychotherapy context, and designs that follow the participants throughout their whole education, make use of other kinds of psychotherapy education and that use matched controls.

## ACKNOWLEDGEMENTS

We would like to thank all research assistants and interns that helped us conduct this study and the clinical psychology students that participated in it. Further, we are very grateful to the reviewers and the editor of this manuscript for their valuable comments and suggestions.

### Funding

The research was supported by a research grant by the Marcus and Amalia Wallenberg Foundation (Marcus och Amalia Wallenbergs Minnesfond; grant no. MAW 2013.0130) and a research grant co-funded by Forte and the Marie Skłodowska-Curie Actions from the EU commission (Cofas FIIP-project, dnr 2013-02727). Stockholm University provided

the open access fees. The funders had no role in study design, data collection and analysis, decision to publish, or preparation of the manuscript.

## Grant Disclosures

The following grant information was disclosed by the authors:
Marcus och Amalia Wallenbergs Minnesfond: MAW 2013.0130.
EU commission: COFAS FIIP-Project, dnr 2013-02727.
Stockholm University.

## Competing Interests

The authors declare that they have no competing interests.

## Author Contributions

- Lillian Döllinger performed the experiments, analyzed the data, prepared figures and/or tables, authored or reviewed drafts of the article, and approved the final draft.
- Isabelle Letellier conceived and designed the experiments, performed the experiments, authored or reviewed drafts of the article, and approved the final draft.
- Lennart Högman conceived and designed the experiments, authored or reviewed drafts of the article, development of computer task, and approved the final draft.
- Petri Laukka analyzed the data, authored or reviewed drafts of the article, development of computer task, and approved the final draft.
- Håkan Fischer conceived and designed the experiments, authored or reviewed drafts of the article, and approved the final draft.
- Stephan Hau conceived and designed the experiments, authored or reviewed drafts of the article, and approved the final draft.

## Human Ethics

The following information was supplied relating to ethical approvals (*i.e.*, approving body and any reference numbers):

The study was approved by the Swedish Ethical Review Authority (Etikprövningsmyndigheten, dnr 2014/1265-31).

## Data Availability

The data is available at OSF: Döllinger, Lillian. 2023. "Open Data Manuscript PeerJ." OSF. August 31. osf.io/rnc3p.

## Supplemental Information

Supplemental information for this article can be found online at http://dx.doi.org/10.7717/peerj.16235#supplemental-information.

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
