# Peer review of "Trainee psychotherapists’ emotion recognition accuracy during 1.5 years of psychotherapy education compared to a control group: no improvement after psychotherapy training"

_PeerJ, doi:10.7717/peerj.16235_

## Round 0.1 · original submission · Minor Revisions

I commed the authors on an interesting piece of work.

Two reviewers have provided feedback for your consideration to help improve your paper. Based on that feedback, I invite you to make some revisions, and provide rationale for changes you do not agree with and wish to keep as is. In your response document please provide a point-by-point response to reviewer suggestions. In that response document if you could provide quotes and pages numbers from the revised article to help make the review process quicker/easier that would be appreciated.

I ask that you consider all the reviewer suggestions. Below are some of my additional thoughts on some of the reviewer comments.

R1 has picked up that some of the conclusions in your initial manuscript are perhaps framed in language that is too strongly worded considering you are reporting on a single study. I agree with those sentiments. For example, in your abstract the statement is too strongly worded: “These results suggest that psychotherapy education and clinical training do not contribute to improved emotion recognition accuracy beyond what could be expected due to time or other factors.” I recommend inserting the word "always" within the statement to read "...clinical training do not always contribute..." to provide some hedging to your conclusions. This is because even though you found no improvement in emotion recognition accuracy in your study, this does not necessitate that this finding would always occur. I recommend revision of some of the other similar statements in your discussion as per R1 suggestions to provide more realistic contained conclusions from your study.

R1 has requested that you write about face masks in the introduction. I can see how perhaps you might include a little bit about this in your discussion section as a point of interest, however I find it hard to see the central relevance of this for your specific work (unless in your work the clients were wearing face masks). Therefore, I don't perceive this as a necessary inclusion into your introduction if you did not wish to not follow this advice. I note that R2 has not mentioned anything on this.

R2 has asked for a 2x2 ANOVA for one of the analyses. While I can see their point (as a matter of personal taste I prefer to use the simplest form of analysis possible to answer research questions to help be as clear and compelling as possible), I do also acknowledge that regression and ANOVA are both linear model analyses that are essentially the same thing, so either approach is technically correct.

Reviewer 1 ·

Basic reporting

The topic is relevant to the journal and the community of clinicians, also considering the critical role of facemasks in emotion recognition. The manuscript is well-written and well-structured. There are some concerns that lead me to suggest a revision of the manuscript.

The introduction describes the topic adequately; however, considering the growing literature on emotion recognition with/without facemasks (also among psychotherapists), I suggest adding a paragraph summarizing these results (e.g., Mitzkovitz, C., Dowd, S.M., Cothran, T. et al. The Eyes Have It: Psychotherapy in the Era of Masks. J Clin Psychol Med Settings 29, 886–897 (2022). https://doi.org/10.1007/s10880-022-09856-x; Bani, M., Ardenghi, S., Rampoldi, G., Russo, S., & Strepparava, M. G. (2023). Impact of facemasks on psychotherapy: Clinician's confidence and emotion recognition. Journal of clinical psychology, 79(4), 1178–1191. https://doi.org/10.1002/jclp.23468; Dalkiran, M. , Yuksek, E. , & Karamustafalioglu, O. (2017). Facial emotion recognition ability in psychiatrists, psychologists, and psychological counselors. European Psychiatry, 41(S1), S157, for example, but there are many other references).
Line 209-212 "At pretest, all participants also filled in a questionnaire about adult attachment and the trainee psychotherapists also answered questionnaires about attitudes regarding psychotherapy, empathy, compassion, and self-compassion": it is unusual for me to read about the constructs measured in the study but see the questionnaires description and results reported as supplementary material. What is the rationale for including attachment, empathy, and self-compassion? Do authors have any hypotheses to explain their inclusion? If these constructs are relevant to the study, please include them in the methods, results, and discussion.

In the study's open-access dataset, I see only ERAM data but not all the other measures included (IRI, self-compassion…)

Considering the declared hypotheses, there are too many tables and figures; I appreciate the transparency of authors adding all the analyses performed, but most of them are not relevant. Limit the number of tables and figures to those needed to test your hypotheses.

Experimental design

no comments

Validity of the findings

Authors wrote: "These results suggest that psychotherapy education and clinical training do not contribute to improved emotion recognition accuracy beyond what could be expected due to time or other factors.", however, without a description of the contents of the psychotherapy training, is difficult to generalize these results to every training.
In the discussion, authors comment on the study's main results, but there are only 4-5 references of previous studies with which to make a comparison; I suggest to comment the results and also considering similar studies.
Authors wrote: "If we assume that ERA does not improve during the practical psychotherapy education, as suggested by our results, and that it is an important therapist skill that is likely positively related to psychotherapy outcome (Abargil & Tishby, 2021), the selection of trainees with high ERA should be encouraged". This implication can be problematic; considering the sample size and the dropout rate, a practical implication that excludes prospective students based on ERA performance seems unethical. In a real clinical context, emotion recognition relies on many verbal and nonverbal information, and patients' features can influence emotion recognition. Suggesting some additional/preliminary training in emotion recognition can be more acceptable.
Among limitations, authors should consider the length of the training; in some countries, psychotherapy training lasts for 2, 3, or 4 years, and we cannot exclude that a change in ERA occurs only later (or after more supervised clinical activities, etc.)

Additional comments

Abstract: Please detail the acronym ERA the first time it appears in the text.
Line 114-126: Please move this section among methods, not after hypotheses.
Lines 348-351, 365-368, and 404-409: There is a duplication of statistical information reported both in the text and table 2; avoid redundancy of information

Sometimes tables and figures report the same information; I appreciate the detailed approach of the authors, but for the readers can be confusing to have too much information. To show the baseline differences between groups and the lack of differences in the follow up use tables or figures not both.

Reviewer 2 ·

Basic reporting

Clear and unambiguous, professional English is used throughout.
o The writers (subliminally) connect emotion recognition and empathy. Although this link might be somewhat intuitive, emotion recognition and empathy are not synonymous, and the connection should be explained.
o I wonder if more theories on emotion recognition (and maybe empathy) highlighting the importance of this function in the psychotherapy relationship would be of impotence to put this research into a theoretical context. I wonder if developmental ideas of parent-infant emotion recognition would help heighten this paper’s significance and relevance.

Experimental design

- Materials & Methods:
o Data Analysis: Major Comment - regarding hypothesis 2 –I don’t think the correct analysis was used. Regression through 2 points is not really “regression” since the line must pass through these two points. I believe a 2X2 ANOVA analysis with group X time would be the better (correct) approach.

Validity of the findings

- Discussion:
o Hypothesis 1: a general comment: make the passage more focused and briefer. What prior research do these results echo or contradict? What does it mean theoretically?
o Hypothesis 2: The null finding may result from not running the correct analysis. I would like to see the results after the ANOVA analysis suggested.

---

## Round 0.2 · accepted · Accept

Thank you for your thorough response to the reviewer feedback.

I believe the manuscript has been improved (although was already in a good place to begin with).

Congratulations on your interesting work.